# Effects of Temperature, Metal Ions and Biosurfactants on Interaction Mechanism between Caffeic Acid Phenethyl Ester and Hemoglobin

**DOI:** 10.3390/molecules28083440

**Published:** 2023-04-13

**Authors:** Yutong Li, Zhen Zhao, Xiao Nai, Mingyuan Li, Jing Kong, Yanrong Chen, Min Liu, Qian Zhang, Jie Liu, Hui Yan

**Affiliations:** 1School of Chemistry and Chemical Engineering, Liaocheng University, Liaocheng 252059, China; 2College of Pharmacy, Liaocheng University, Liaocheng 252059, China

**Keywords:** caffeic acid phenethyl ester, hemoglobin, bile salt, temperature, metal ion

## Abstract

Caffeic acid phenylethyl ester (CAPE) is a natural polyphenol extracted from propolis, which is reported to have several pharmacological effects such as antibacterial, antitumor, antioxidant and anti-inflammatory activities. Hemoglobin (Hb) is closely related to the transport of drugs, and some drugs, including CAPE, can lead to a change in Hb concentration. Herein, the effects of temperature, metal ions and biosurfactants on the interaction between CAPE and Hb were studied using ultraviolet-visible spectroscopy (UV−Vis), fluorescence spectroscopy, circular dichroism (CD), dynamic light scattering (DLS) and molecular docking analysis. The results showed that the addition of CAPE led to changes in the microenvironment of Hb amino acid residues as well as the secondary structure of Hb. Hydrogen bonding and van der Waals force were found to be the main driving forces for the interaction between CAPE and Hb through fluorescence spectroscopy and thermodynamic parameter data. The results of fluorescence spectroscopy also showed that lowering the temperature, adding biosurfactants (sodium cholate (NaC) and sodium deoxycholate (NaDC)) and the presence of Cu^2+^ increased the binding force between CAPE and Hb. These results provide useful data for the targeted delivery and absorption of CAPE and other drugs.

## 1. Introduction

Caffeic acid phenylethyl ester (CAPE; its structure is shown in Figure 1), the biologically active constituent of propolis, is of great interest due to its role in human health and diseases. At present, CAPE has been studied in the treatment of oral cancer, prostate cancer, intestinal cancer and other cancers [1,2,3]. In addition to its strong biotic properties, such as its antibacterial, antiviral, anti-inflammatory, antioxidative, antiplatelet and antitumor activities, CAPE also has certain potential in the treatment of infections, diabetes, neurodegeneration, anxiety and cardiovascular diseases [4,5]. Hassan et al. found that CAPE can inhibit the increase in serum TNF-α level, induce HO-1 aortic expression and reduce collagen deposition. Moreover, CAPE was found to eliminate the symptoms of diabetes-induced atherosclerosis without any effect on the pre-existing symptoms of hyperglycemia. These results suggest that CAPE may be an important drug to protect the vasculature in diabetic patients [6]. Kuo et al. found that CAPE can reduce the secretion of vascular endothelial growth factor (VEGF) by inhibiting the ROS, PI3K and HIF-1α signaling pathways in human retinal pigment epithelial cells under hypoxic conditions, thereby inhibiting tumor growth and metastasis [7,8]. Kumar et al. carried out research on the role of CAPE in dementia and found that by increasing the level of glutathione (GSH) in the brain of rats and enhancing antioxidant protection, memory can be improved and memory impairment can be prevented [9]. It was also reported that CAPE may directly target the signaling pathways related to oxidation, inflammation and cancer development, such as NF-κB (nuclear factor-κB), MAPK (mitogen-activated protein kinase), PI3K/PKB (phosphoinositide 3-kinase/protein kinase B) and NLPR3 (NOD-like receptor thermal protein domain associated protein 3). As a result, it can combat the oxidative stress, inflammation, proliferation, migration and invasion of cells [3,4,10]. The above studies show that CAPE is an important target drug for the prevention and treatment of cardiovascular, neurological and cancer-related diseases. However, the current research results of CAPE are mainly from cellular and animal model experiments. Studies from the aspects of pharmacodynamics and pharmacokinetics promote the further development and clinical transformation of CAPE. Therefore, our research group previously used pepsin, trypsin and *α*-chymotrypsin as digestive enzyme models to study the interactions between CAPE and proteases [11]. The effects of temperature, pH, metal ions and biosurfactants on the binding of CAPE to BSA were also studied [12]. The research results addressed the deficiencies of related studies and provided relevant thermodynamic data.

Currently, there is considerable research on the intermolecular interactions between proteins and many ligands [13,14,15,16]. However, studies on the interaction between drugs and hemoglobin (Hb) are relatively few. As the most important component of red blood cells, with a molecular weight of 64.45 kDa, Hb transports oxygen and carbon dioxide in the blood vessels of animals and regulates blood pH [17,18,19]. When drugs enter the blood, Hb can interact with small drug molecules, thereby reducing the concentration of free drugs and affecting the distribution, transport and activity of drugs. This special physiological function is closely related to the unique molecular structure of Hb [20]. Therefore, in recent years, Hb has been widely used as a model protein for studying the interactions between active small molecules and biomacromolecules [21]. Bovine hemoglobin has 90% amino acid sequence homology with human hemoglobin, so bovine hemoglobin was used instead of human hemoglobin in this study. In the research on Hb, some influencing factors of the Hb–drug interaction need to be actively considered. First, biosurfactants may alter the binding behavior of drugs to proteins to regulate drug release. Thus, the bile salts (BSs) biosurfactants, including sodium deoxycholate (NaDC) and sodium cholate (NaC), were chosen as the influencing factor. BSs are naturally occurring amphiphilic substances that are synthesized and released by the liver and stored in the gallbladder, and their effect on the interaction between proteins and ligands is often studied [22]. Second, temperature and metal ions were also chosen as the influencing factors. Therefore, in this study, the quenching mode, binding affinity and interaction force of the Hb–CAPE system were investigated by fluorescence spectroscopy, and the structural changes in Hb were analyzed by circular dichroism (CD) spectroscopy, ultraviolet absorption (UV−Vis) spectroscopy and dynamic light scattering (DLS). The effects of temperature, biosurfactants and metal ions on the interaction between Hb and CAPE were also studied. In addition, molecular docking can offer theoretical evidence for the type of force between Hb and CAPE. The obtained experimental data and theoretical model can provide important data on the in vitro binding reaction, mode of action and structural specificity of CAPE and Hb. This can lay the foundation for further research on the digestion, transport and absorption of drugs in the human body.

## 2. Results and Discussion

### 2.1. Effect of Temperature on the Binding Interaction of CAPE–Hb

#### 2.1.1. Quenching Mechanism and Thermodynamic Analysis

The fluorescence emission spectra of Hb and CAPE–Hb system at 298.2 K and pH 7.4 are presented in Figure 2a. As seen from the figure, under the excitation wavelength of 280 nm, Hb exhibited a fluorescence emission peak around 335 nm. With the increase in CAPE concentration, the intensity of the Hb fluorescence peak decreased, but the position and the shape of the maximum absorption peak did not change. The fluorescence of Hb was quenched to varying degrees by the formed CAPE–Hb complexes, and the quenching mode can be judged based on the presence of dynamic or static quenching [23]. Dynamic quenching refers to fluorescent quenching due to the transfer of energy or physical collision between the excited fluorescent molecule and the quencher, while static quenching is due to the formed complex between the ground state fluorescent molecule and the quencher by weak bonds. It is reported that the dynamic quenching constant increases with temperature, while the static quenching constant follows the opposite trend [24]. According to this empirical rule, the Stern–Volmer equation can be used to evaluate the quenching mechanism of Hb and CAPE [23]:(1)F0F=Ksv[Q]+1=Kqτ0[Q]+1
where *F*_0_ and *F* represent the fluorescence intensity of Hb in the absence or presence of CAPE, respectively; *K_sv_* is the Stern–Volmer quenching constant; *K_q_* is the quenching rate constant; [*Q*] is the total concentration of CAPE; and $\tau_{0}$ is the average fluorescence lifetime of the protein (the fluorescence decay lifetime is usually 10^−8^ s) [25]. Considering the inner filter effects of CAPE on the fluorescence spectra of Hb, the fluorescence intensities in this experiment were corrected by the following equation:(2)$$F_{cor}=F_{obs}\exp[\left(A_{ex}+A_{em}\right)/2]$$
where *F_cor_*, *F_obs_*, *A_ex_* and *A_em_* are the corrected fluorescence intensity, the observed fluorescence intensity, and the absorbance values of the sample at excitation and emission wavelengths, respectively. The fluorescence quenching constants of the interaction between CAPE and Hb at four temperatures can be obtained from the slopes of the Stern–Volmer plots shown in Figure 2b. The obtained data are shown in Table 1. The *K_q_* values of the CAPE–Hb system in pH 7.4 phosphate buffer solution were 6.41 × 10^12^ M^−1^·s^−1^, 6.00 × 10^12^ M^−1^·s^−1^, 5.81 × 10^12^ M^−1^·s^−1^ and 5.45 × 10^12^ M^−1^·s^−1^ at the temperatures of 298.2 K, 302.2 K, 306.2 K and 310.2 K respectively. Therefore, the quenching mechanism between CAPE and Hb was static quenching.

Based on the premise of static quenching between the protein and the ligand, the binding constant (*K_a_*) and the number of binding sites (*n*) can be calculated according to the following double logarithmic formula [26]:(3)$$\log(F_{0}/F-1)=n\log{\left[Q\right]}_{0}+\log K_{a}$$
where [*Q*]_0_ is the free concentration of the drug, which can be calculated by the following formula, and [*P*]_0_ is the total concentration of protein:(4)$${\left[Q\right]}_{0}=[Q]+n(F_{0}-F){\left[P\right]}_{0}/F_{0}$$

At different temperatures, the double-logarithmic curves of the interaction between CAPE and Hb showed a positive linear correlation, as shown in Figure 2c. The obtained *K_a_* and *n* values from the intercept and slope are shown in Table 1. In the temperature range studied, the binding constant decreased, and the binding site did not change significantly with the increase in temperature. The results show that high temperature reduced the binding affinity of CAPE to Hb, which was not conducive to the binding of CAPE to Hb.

The type of force involved in the process of ligand–protein binding can be judged by thermodynamic parameters (∆*H*_m_, ∆*S*_m_ and ∆*G*_m_) [27]. ∆*H*_m_ and ∆*S*_m_ can be treated as constants if they do not vary much over the temperature range. In order to obtain the thermodynamic parameters of the interaction between CAPE and Hb (shown in Table 1), the following formulas are used:(5)$$\ln K_{a}=-\frac{\Delta H_{m}}{RT}+\frac{\Delta S_{m}}{R}$$
(6)$$\Delta G_{m}=-RT\ln K_{a}=\Delta H_{m}-T\Delta S_{m}$$

Equation (5) is the Van’t Hoff formula, and the values of ∆*H*_m_ and ∆*S*_m_ can be obtained from the slope and intercept of the plot of ln*K_a_* against 1/*T* (Figure 2d). Then, Equation (6) was used to calculate ∆*G*_m_. According to the theoretical analysis of Ross and Subramanian [28], both negative ∆*H*_m_ and ∆*S*_m_ values mean that hydrogen bond and van der Waals force are the main forces between the ligand and the protein. This may be related to the strong polar phenolic hydroxyl groups in CAPE, which can form hydrogen bonds with amino acids having hydroxyl groups in their side chains. A negative value of ∆*G*_m_ indicates that the thermodynamic process of CAPE binding to Hb is spontaneous.

#### 2.1.2. Time-Resolved Fluorescence Spectrometry

Time-resolved fluorescence is very sensitive to the excitation response and is the best way to distinguish quenching modes [29]. In order to directly determine the quenching mechanism between Hb and CAPE, the lifetime of the excited state was measured by dynamic fluorescence spectroscopy. The measured fluorescence decay curve was fitted using a double exponential, and the average lifetime of Hb was calculated using the following formula [30]:(7)$$\tau =\alpha_{1}\cdot \tau_{1}+\alpha_{2}\cdot \tau_{2}$$
where *τ* is the average fluorescence lifetime and *α*_1_ and *α*_2_ are the pre-exponential factors for the first and second decay times *τ*_1_ and *τ*_2_.

The fluorescence lifetime data of Hb in the presence or absence of CAPE at 298.2 K are shown in Table 2. The fluorescence lifetime *τ* of Hb did not change significantly with the addition of CAPE or changes in its concentration. These results further indicate that the quenching mechanism between CAPE and Hb was static quenching, which was consistent with the conclusions of fluorescence quenching spectroscopy.

#### 2.1.3. UV−Vis Spectrum Analysis

UV−Vis spectroscopy can be used to detect changes in the microenvironment of aromatic amino acid residues in proteins. The spectra of the CAPE–Hb system at 298.2 K and pH 7.4 are shown in Figure 3a. In the absence of CAPE, Hb displayed two absorption bands, located at 278 nm and 410 nm. The absorption peak at 278 nm is the characteristic absorption peak of amino acid residues (Trp and Tyr), and the absorption peak at 410 nm is the characteristic absorption peak of heme moiety [18]. After adding CAPE, an absorption band appeared at 324 nm, which is the characteristic absorption band of polyphenolic compounds. With the increase in CAPE concentration, the intensity of the absorption peak located at 278 nm continuously increased, and a red shift appeared. This indicated that CAPE interacted with the aromatic residues of Hb, thereby affecting the microenvironment of the Hb amino acid residues, which increased the hydrophobicity and decreased the polarity of the aromatic amino acid residues and changed the tertiary structure of Hb. However, when the concentration of CAPE increased, the absorption peak located at 410 nm decreased, but the change was not obvious. This suggests that the stability of heme was not significantly affected, although the interaction between CAPE and Hb induced a change in the pocket conformation of the protein, as confirmed by the fluorescence method and the subsequent molecular docking analysis.

#### 2.1.4. Secondary Structure Analysis

In order to study the change in Hb’s structure after its combination with CAPE, the CD spectrum of Hb was measured in the presence or absence of CAPE, as shown in Figure 3b. The protein of Hb exhibited obvious negative bands near 208 nm and 220 nm [21]. The mean residual ellipticity (MRE, in deg·cm^2^/dmol) can be obtained from Equation (8). In the absence of CAPE, the molar ellipticity of Hb was relatively small. After adding different amounts of CAPE (ratios of Hb to CAPE were 1:5, 1:20 and 1:30), the MRE value at 208 nm increased by about 2.9%, 8.1% and 12.8%, and the MRE value at 220 nm increased by about 2.2%, 7.0% and 9.6%, respectively. This confirmed the binding of CAPE to Hb and reflected the conformational change in the secondary structure of Hb [31]. The peak position of the spectral line did not change, but the spectral line moved upward, indicating that the addition of CAPE led to a constant decrease in *α*-helix and promoted the transformation of *α*-helix into other secondary structures. These data are consistent with the results of UV−Vis spectroscopy:(8)$$MRE(\deg\cdot {\mathrm{cm}}^{2}/\mathrm{dmol})=\frac{Intensity\;of\;CD(\mathrm{mdge})\times 1000}{C_{p}(\mathrm{mmol}/L)\times n\times l(\mathrm{mm})}$$
where *C_p_* is the concentration of the protein (5.0 × 10^−3^ mM), *n* is the number of amino acid residues in the protein (572 for Hb) and *l* is the length of the light path (10 mm).

#### 2.1.5. Particle Size Analysis

The DLS method can measure the hydrodynamic diameter (*v*) of biomolecules in solution [32] and rapidly characterize the size of molecules [33]. Figure 3c presents the dynamic light scatter diagram of the interaction between CAPE and Hb at 298.2 K and in pH 7.4 phosphate buffer solution. The corresponding *D*_h_ data of Hb are listed in Figure 3d. With the increase in CAPE concentration, the dynamic scattering light intensity of Hb gradually decreased, which indicated that CAPE and Hb formed a complex. In the absence of CAPE, the *D*_h_ of Hb was 7.41 nm. With the increase in CAPE dosage, the *D*_h_ of Hb decreased continuously. The interaction between CAPE and Hb did not necessarily lead to the unfolding and relaxation of the protein, but the transformation of its secondary structure was confirmed using the DLS method. The experimental results were the same as the conclusions of CD and UV-Vis spectroscopy.

### 2.2. Effect of Biosurfactants (NaC, NaDC) on the Binding Interaction of CAPE–Hb

As a series of naturally biocompatible biosurfactants [34,35], bile salts can be dissolved by cholesterol, lipids and other non-polar substances. Hence, they play an important physiological role in the digestion and absorption processes for insoluble drugs. Understanding the interaction mechanism between bile salts and CAPE–Hb complexes can help to evaluate the in vitro metabolism of drugs.

The effect of two biosurfactants (NaC, NaDC) on the fluorescence spectrum of Hb is shown in Figure 4a. In the presence of surfactants, the peak position and peak shape of Hb did not change, but the fluorescence intensity decreased significantly, indicating that the addition of biosurfactants induced the exposure of the amino acid residues of Hb (Trp or Typ) and thus changed the microenvironment of the amino acid residues in the protein. Then, the effect of biosurfactant concentration on the CAPE–Hb interaction was studied. The reported values of the critical micelle concentration (CMC) of NaC and NaDC measured by different methods are 5~13 mM and 2~70 mM, respectively, and the pre-CMC/CMC values are 3.4/9.9 mM and 2.5/7.3 mM for NaC and NaDC, as determined by isothermal titration microcalorimetry (ITC) [11]. Based on these data, four biosurfactant concentrations were selected for further research, and the results are listed in Table 3. As seen from the fluorescence quenching spectra of the interaction between CAPE and Hb in the presence of NaC and NaDC in Figure 4b,c, the peak position and peak shape of Hb did not change significantly, but the fluorescence intensity decreased regularly with the increase in CAPE concentration. Using the Stern–Volmer equation to fit the data (insets in Figure 4b,c), it can be seen that there was a good linear relationship between *F*_0_/*F* and CAPE concentration. This result means that the surfactants did not have an impact on the quenching effect of CAPE on Hb, and the quenching mechanism of CAPE to Hb was still a single static quenching mechanism after adding a surfactant. Equations (2)–(5) were used to process the data for the CAPE–Hb system in the presence of surfactants (NaC, NaDC) in pH 7.4 buffer solution. Analysis of the data in Table 3 indicated that the monomers or micelles of biosurfactants played a certain role in protecting the combination of CAPE and Hb through the unfolding of hydrophobic amino acids. The binding constant (*K_a_*) was increased by one or two orders of magnitude with the addition of NaC or NaDC. The surfactant obviously increased the binding affinity between CAPE and Hb, which may be due to the surfactant changing the microenvironment of Hb amino acids and opening the hydrophobic cavity. Micelle formation is unfavorable to the combination of CAPE and Hb. It is likely that the distribution of CAPE in the micelle microenvironment prevented the combination of Hb and CAPE.

Excess micelle formation increases the complex impact on the interaction of CAPE and Hb. Thus, the concentrations of NaC and NaDC were fixed at 3.3 mM and 2.5 mM with the highest possible concentration around the pre-CMC to study the effect of temperature on the combination of CAPE and Hb. The calculated quenching constants, binding constants, number of binding sites and thermodynamic parameters are listed in Table 4. It was found that the quenching constant decreased with the increase in temperature in the presence of NaC or NaDC, indicating that the quenching mechanism of CAPE to Hb was still static quenching and that the surfactant had no significant effect on the quenching mechanism. As shown in Table 4, in the presence or absence of biosurfactants, the thermodynamic parameters (∆*H*_m_, ∆*S*_m_ and ∆*G*_m_) were all negative. Hence, the formation of CAPE–Hb complexes was a spontaneous enthalpy-driven process, co-driven by hydrogen bonding and van der Waals force. In brief, the surfactants (NaC, NaDC) had no significant effect on the driving force and interacting mode between CAPE and Hb but did strongly enhance the binding strength between them. Thus, the addition of a biosurfactant can decrease the free drug concentration, help achieved sustained drug release and improve efficacy.

### 2.3. Effect of Metal Ions on Binding Interaction of CAPE–Hb

After metal ions enter living organisms, they can interact with proteins and change the structure and function of those proteins. Thus, metal ions play an important role in the binding interaction between drugs and metalloproteins. The research on the effect of metal ions on CAPE–Hb complexes can help to regulate the in vivo metabolism of drugs. Different metal ions have different impacts on the interactions between drugs and proteins, sometimes inducing an enhanced effect, such as the systems of (Mg^2+^, Mn^2+^, Co^2+^, Fe^2+^, Zn^2+^, Ca^2+^, Cu^2+^ or Ag^+^)−baicalein−BSA [36] and Cu^2+^−(galangin or baicalin)−BSA [37]. Sometimes, they induce a weakened effect, such as in the systems of (Cu^2+^ or Fe^3+^)−baicalein−BSA [38] and Cu^2+^−(chrysin, baicalein, luteolin, or vitexin)−BSA [37]. In order to investigate the competitive or cooperative effect of metal ions on CAPE–Hb interaction, the binding constants (*K_a_*) of CAPE and Hb in the presence of different metal ions (K^+^, Cu^2+^, Ca^2+^, Ni^2+^, Mn^2+^) at 298.2 K were determined using fluorescence spectroscopy. These metal ions are essential life elements involved in various human metabolism processes, whose concentrations were fixed at 60 μM considering their trace existence in the human body.

Figure 5 shows the double logarithmic curve of the CAPE–Hb system in the presence of metal ions in pH 7.4 buffer solution. Then, the *K_a_* value can be obtained from the intercept using Equation (2), as listed in Table 5. The presence of metal ions (K^+^, Ca^2+^, Mn^2+^) decreased the *K_a_* values to varying degrees, especially Ca^2+^. The decreased binding affinity between CAPE and Hb may be due to the competition between metal ions and CAPE for Hb, that is, the formation of metal ion–Hb complexes may affect the conformation of Hb and inhibit the combination of CAPE and Hb [39,40]. Some oral drugs are not recommended to be taken along with calcium, because it greatly affects the delivery and metabolism of drugs. The presence of Ni^2+^ slightly strengthened the CAPE–Hb interaction, but moderate supplementation with nickel can increase the production of red blood cells and Hb in the body, which is an important applied factor to be considered in vivo. Finally, the presence of Cu^2+^ significantly increased the binding constant of CAPE and Hb. It is known that both iron and copper play important roles in hematopoiesis. Ceruloplasmin can catalyze the oxidation of Fe^2+^, thus promoting the binding of Fe^2+^ and ferroprotein to facilitate iron transport. Therefore, Cu^2+^ likely reacts with CAPE to form a metal ion–CAPE complex [39,41]. The metal ion–CAPE complex interacts with Hb to form a new metal ion–CAPE–Hb complex, where the presence of Cu^2+^, acting as a “bridge”, leads to an obvious increase in the binding strength of CAPE and Hb.

### 2.4. Molecular Docking

The molecular docking method can be used to validate the experimental results of spectroscopy to a certain extent [42]. In order to study the effect of the type of Hb and pH on the simulation diagram of Hb docking with CAPE, NaC and NaDC, carbonmonoxy liganded bovine Hb (PDB ID: 1G08, 1G09 and 1G0A at pH 5.0, 7.2 and 8.5), bovine deoxy Hb (PDB ID: 1HDA) and the CO form of bovine Hb (PDB ID: 6II1) were chosen. The bovine Hb of 1G09 at pH 7.2 was taken as an example to analyze, and the results of the other systems are shown in Appendix A. For the Hb and CAPE/NaC/NaDC system, the amino acid residues lining the binding site in the Hb cavity, the hydrogen bonds and the lowest binding energy are summarized in Appendix A.

The simulation diagrams of CAPE, NaC and NaDC in Hb are shown in Appendix A, and their surrounding amino acid residues are shown in Figure 6. As shown in Figure 6a, CAPE is bound inside the Hb helix. Among the amino acid residues, THR134, THR137, LYS104, LYS99, SER133, ASN108, TYR35 and HIS103 are hydrophilic amino acid residues with very strong polarities, which can form van der Waals forces with the polar bonds of CAPE molecules. In addition, ASN108 and HIS103 form two hydrogen bonds and one hydrogen bond with the hydroxyl group in the CAPE molecule, respectively, and SER133 forms one hydrogen bond with the carbonyl group. Their distances are 1.90 Å, 2.11 Å, 2.71 Å and 1.65 Å, respectively. As shown in Figure 6b, NaC is also bound inside the Hb helix. The amino acid residues of THR137, THR134, TYR140 and SER138 each form one hydrogen bond with the hydroxyl group in the NaC molecule. LYS99 forms two hydrogen bonds with the carbonyl group. Their distances are 1.73 Å, 2.16 Å, 1.66 Å, 1.86 Å, 1.73 Å and 1.92 Å, respectively. The interactions of the amino acid residues of NaDC with Hb in Figure 6c are the same as those of NaC. The difference is that the distances of these hydrogen bonds are 1.87 Å, 2.27 Å, 1.62 Å, 1.87 Å, 1.73 Å and 1.96 Å, respectively. The results of molecular docking show that the interaction forces between Hb and the three molecules (CAPE, NaC and NaDC) all include hydrogen bonds and van der Waals force, which is consistent with the conclusions obtained from fluorescence spectroscopy.

The docking positions of small molecules in Hb as a whole were compared and analyzed (Figure 6d,e). The similar positions of NaC and NaDC and their close position to CAPE may help to explain the effect of the biosurfactant on the binding action of CAPE–Hb. The addition of biosurfactants not only results in exposure of the amino acid residues of Hb, but also enhances the binding strength through the interaction force between CAPE and the biosurfactant. In addition, the lowest binding energies of Hb interaction with CAPE, NaC and NaDC obtained by molecular docking were −32.3 kJ/mol, −40.9 kJ/mol and −41.1 kJ/mol, respectively. This showed that the combinations were spontaneous and the binding tendency of biosurfactant on Hb was stronger than CAPE. The different binding positions of biosurfactants and CAPE verify the results of fluorescence spectroscopy. The strong combination of surfactant (NaC, NaDC) and Hb induced the change in the tertiary structure of the protein. However, the surfactant did not occupy the position of CAPE; on the contrary, the binding interaction between CAPE and Hb was strengthened.

Using a similar analysis, several points can be obtained based on the data in Appendix A. (i) The surrounding amino acid residues of NaDC and the formed hydrogen bonds with the same binding protein are similar as those of NaC. The absolute value of the lowest binding energy follows the order of CAPE–Hb < NaC–Hb < NaDC–Hb. (ii) Among the different types of Hb, the bovine deoxy Hb has the strongest interaction with the small molecules, then the CO form of bovine Hb and the carbonmonoxy liganded bovine Hb. (iii) A basic environment (pH 8.5) helps to enhance the interaction between Hb and CAPE/NaC/NaDC, and acidic (pH 5.0) and neutral (pH 7.2) environments have no significant impact on the interaction between them. These results help to play a regulatory effect on drug targeting and slow release action.

## 3. Materials and Methods

### 3.1. Materials

The purity levels and sources of the chemicals used in this study are listed in Table 6. CAPE is a mixture of cis−isomer and trans-isomer, whose contents are 13.46% and 86.54%, respectively, and were determined by high-performance liquid chromatography (HPLC). All chemicals were used as received without further purification. In this study, phosphate buffer was prepared from Na_2_HPO_4_ and NaH_2_PO_4_ in a certain ratio using triple-distilled water with a specific conductivity (below 3 μs·cm^−1^). All test solutions were diluted with phosphate buffer to the desired volume.

### 3.2. Preparation of Sample

Stock solution of Hb (100 µM) was prepared using a 10 mM phosphate buffer at pH 7.4. The CAPE stock solution with a concentration of 400 µM was prepared with 10 mM pH 7.4 phosphate buffer and ethanol (9:1 buffer: ethanol). Stock solutions of NaC, NaDC and inorganic salts were prepared with 0.1 M pH 7.4 phosphate buffer solution. The concentrations of NaC and NaDC stock solutions were 100 mM, and the concentrations of CaCl_2_, CuCl_2_·2H_2_O, KCl, MnCl_2_·4H_2_O and NiCl_2_·6H_2_O were all 1.5 mM. The quantity of crystallization water in CuCl_2_·2H_2_O, MnCl_2_·4H_2_O or NiCl_2_·6H_2_O was calculated when preparing inorganic salt stock solution.

### 3.3. Fluorescence Quenching Spectroscopy

Fluorescence spectra were measured using a fluorescence spectrometer (F-7100, Hitachi, Japan). The final concentration of Hb was fixed at 5 µM. Appropriate amounts of the stock solutions (100 µM for Hb and 400 µM for CAPE) were taken and diluted to the desired concentrations with phosphate buffer. The solutions were kept at 298.2 K, 302.2 K, 306.2 K and 310.2 K for 20 min before measurement. Then, the isothermal solution was placed into a 10 mm quartz cuvette equipped with circulating water with an accuracy of 0.1 K for fluorescence analysis. The fluorescence emission spectra were recorded at 290~450 nm at an excitation wavelength of 280 nm, and the slit widths were 2.5 nm and 5 nm, respectively [43]. The biosurfactant–CAPE–Hb solutions were prepared by adding NaC and NaDC stock solutions to the mixed CAPE–Hb system. The concentrations of NaC and NaDC were 3.3 mM and 2.5 mM. The preparation method of the metal ion–CAPE–Hb system was the same as that of the biosurfactant–CAPE–Hb system, and the test concentrations of all metal ions (K^+^, Cu^2+^, Ca^2+^, Ni^2+^, Mn^2+^) were all 60 μM. The experimental conditions, test methods and instrument parameters were the same as those described above. The results of three scans were averaged, and background fluorescence was corrected.

### 3.4. Time-Resolved Fluorescence Spectrometry

The fluorescence lifetime of Hb in the presence or absence of CAPE was measured using a dynamic fluorescence spectrometer (FLS 1000, Edinburgh Instruments, Livingston, United Kingdom). The concentration of Hb was 5 µM, and the molar ratio of Hb to CAPE varied as 1:0, 1:1, 1:2 and 1:3. The sample recording area of fluorescence decay was 0~100 ns, and the excitation and emission wavelengths were 280 nm and 334 nm. The attenuation data of the samples were fitted and analyzed using the Fluoracle software of the instrument.

### 3.5. UV−Vis Adsorption Spectroscopy (UV−Vis)

The ultraviolet spectra of the interaction between Hb and CAPE were measured using a UV-vis spectrophotometer (UH4150, Hitachi, Tokyo, Japan) with a test range of 180~450 nm and a scanning speed of 700 nm/min. The sample preparation was consistent with that of fluorescence spectroscopy, and the buffer solution was used as the blank solution for background correction.

### 3.6. Circular Dichroism (CD)

The CD spectra were measured at 298.2 K using a CD spectrometer (Jasco-810, Tokyo, Japan) with a special 500 µL quartz cuvette. The spectral scanning range was 180~260 nm, and the scanning speed was 700 nm/min. The same sample was scanned three times in parallel, and the average value was taken. Blank correction was performed during the test. All measurements were carried out under nitrogen atmosphere. The concentration of Hb was fixed at 5 µM, and the molar ratio of Hb to CAPE varied as 1:0, 1:5, 1:20 and 1:30.

### 3.7. Dynamic Light Scattering (DLS)

The hydrodynamic diameter (*D*_h_) of CAPE-Hb complexes with different proportions was measured by a DLS instrument (MRT-2, Malvern, United Kingdom). The concentration of Hb was fixed at 5 µM, and the molar ratio of Hb to CAPE varied as 1:0, 1:1, 1:3 and 1:5. All freshly prepared solutions were filtered through 0.22 μm hydrophilic polyvinylidene difluoride (PVDF) membrane filters several times before measurement. Each sample was tested at least three times, and the test results were directly obtained by ZETASIZER NANO software V4.10.

### 3.8. Molecular Docking

The interactions and binding sites of CAPE to Hb were simulated using AutoDock 4.2.6 semi-flexible docking method. The three-dimensional structure models of the carbonmonoxy liganded bovine Hb (PDB ID: 1G08, 1G09 and 1G0A at pH 5.0, 7.2 and 8.5), the bovine deoxy Hb (PDB ID: 1HDA) and the CO form of bovine Hb (PDB ID: 6II1) were downloaded from the protein database (https://www.rcsb.org/, URL (accessed on 3 April 2023)). Detailed information about the molecular docking process can be found in literature [11,44]. After calculation with AutoGrid 4.0 software, the docking position of the CAPE-Hb (NaC-Hb, NaDC-Hb) complex with the lowest binding energy was obtained, and the visual image of the binding site was generated using VMD1.9.2 software to analyze the type of molecular interaction.

### 3.9. Statistical Analysis

All experiments were repeated three times independently. SPSS software 26.0 was used to carry out *t*-tests and one-way ANOVA. The *p*-value < 0.05 was considered statistically significant.

## 4. Conclusions

Hb is closely related to the transport of drugs, and some drugs can lead to changes in Hb concentration. Therefore, the interaction between CAPE and Hb and their binding mechanism were studied using several methods in this work. The results of fluorescence spectroscopy and thermodynamic parameters indicated that hydrogen bonds and van der Waals force were the main driving forces for the interaction between CAPE and Hb, and high temperature was not conducive to the formation of CAPE–Hb complexes. The quenching mechanism between CAPE and Hb was static quenching, which was also confirmed by the fluorescence lifetime method. CD spectroscopy showed that the addition of CAPE reduced the α-helical structure of Hb and promoted its conversion to other secondary structures. Changes in the microenvironment of Hb amino acid residues and the secondary structure of Hb were also confirmed by UV−Vis spectroscopy and dynamic light scattering. The addition of biosurfactants and metal ions had a regulating effect on the binding strength of CAPE and Hb. The metal ions of Cu^2+^ and Ca^2+^ had a significant effect on the CAPE–Hb interaction; a promoting effect was induced by Cu^2+^ and a decreasing effect was induced by Ca^2+^. These results are significant, as the structure of Hb and the binding strength of Hb-CAPE can be effectively adjusted by different temperatures and additives. This adjustment might be relevant to targeted drug delivery and release of CAPE when employed as injection and oral medication for the treatment of cardiovascular, neurological and cancer-related diseases. The decreased binding strength of Hb–CAPE may be favorable for rapid drug release, and the increased binding strength may favor sustained drug release, thus achieving the desired efficacy. Therefore, the binding study of drugs to Hb is highly important in pharmacology, pharmacokinetics and other fields. In addition, these results provide a theoretical basis and research basis for further studies on the properties of CAPE, sustained release in vivo and anti-tumor mechanisms.

## Figures and Tables

**Figure 1 molecules-28-03440-f001:**
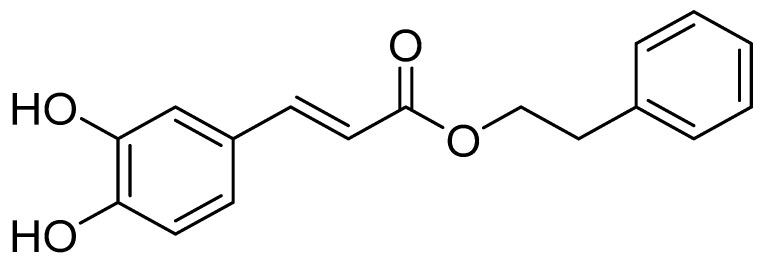
Chemical structure of CAPE.

**Figure 2 molecules-28-03440-f002:**
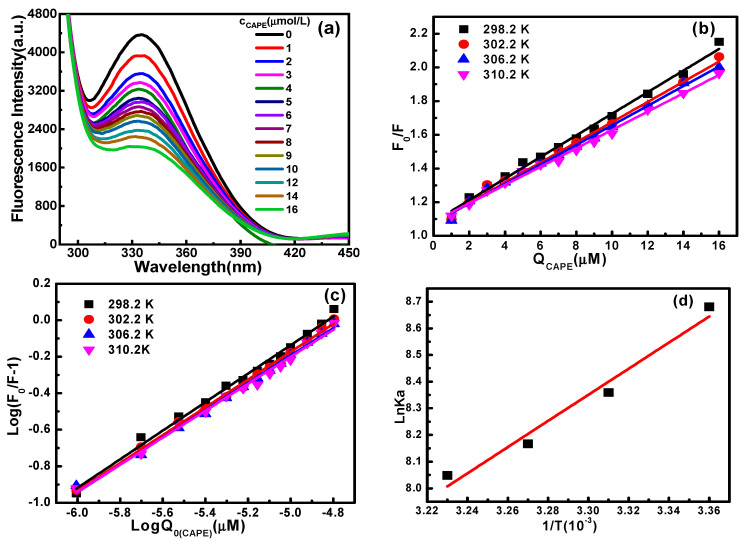
(**a**) Fluorescence spectra at 298.2 K; (**b**) Stern–Volmer plots at different temperatures; (**c**) The log((*F*_0_ − *F*)/*F*) and log[*Q*]_0_ double-logarithmic plots at different temperatures; and (**d**) Van’t Hoff plots of ln*K_a_* versus 1/*T* for the CAPE−Hb system in pH 7.4 phosphate buffer solution.

**Figure 3 molecules-28-03440-f003:**
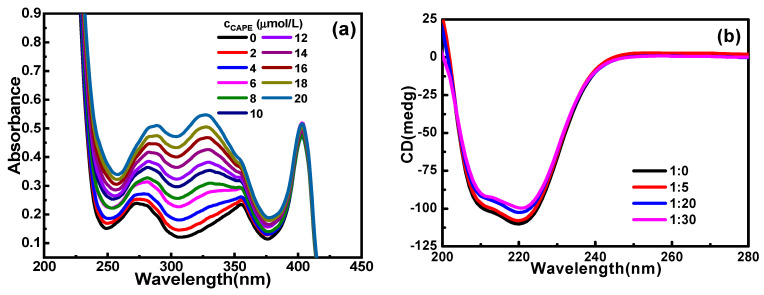
(**a**) UV−Vis absorption spectra; (**b**) CD spectra; (**c**) DLS diagram; and (**d**) *D*_h_ data of Hb with or without CAPE for the CAPE–Hb system at 298.2 K and in pH 7.4 phosphate buffer solution.

**Figure 4 molecules-28-03440-f004:**
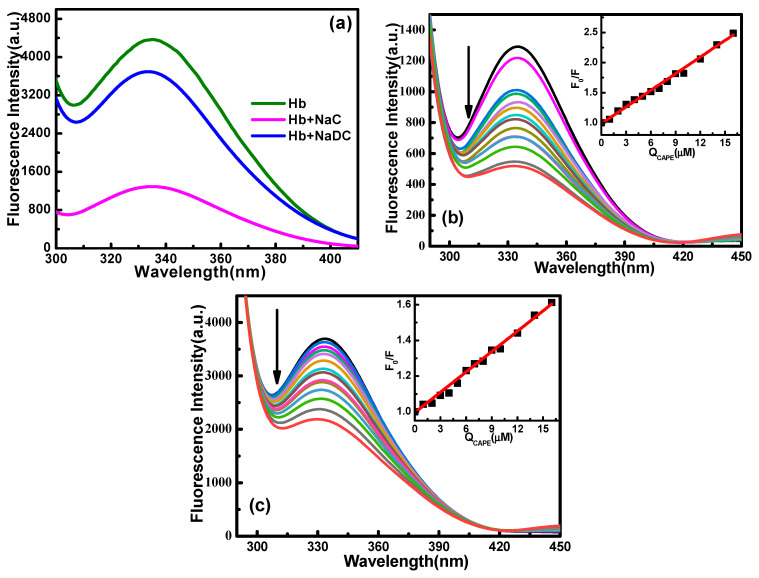
Fluorescence spectra of Hb for different systems at 298.2 K and in pH 7.4 phosphate buffer solution: (**a**) Surfactant–Hb system; (**b**) NaC–CAPE–Hb system; (**c**) NaDC–CAPE–Hb system. The concentrations of NaC and NaDC are fixed at 3.3 mM and 2.5 mM. The concentrations of CAPE are 0, 1, 2, 3, 4, 5, 6, 7, 8, 9, 10, 12, 14 and 16 µM from top to bottom in direction of arrow. The insets in Figure 3b,c show the corresponding Stern–Volmer diagrams.

**Figure 5 molecules-28-03440-f005:**
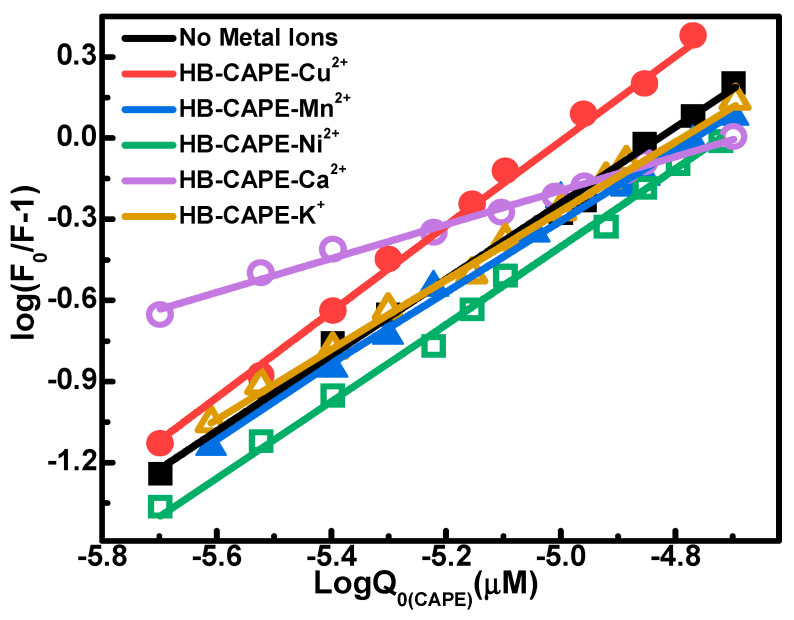
Double−logarithmic diagram of the CAPE–Hb system in the presence of five different metal ions at 298.2 K and in pH 7.4 buffer solution. The concentration of metal ions was fixed at 60 μM.

**Figure 6 molecules-28-03440-f006:**
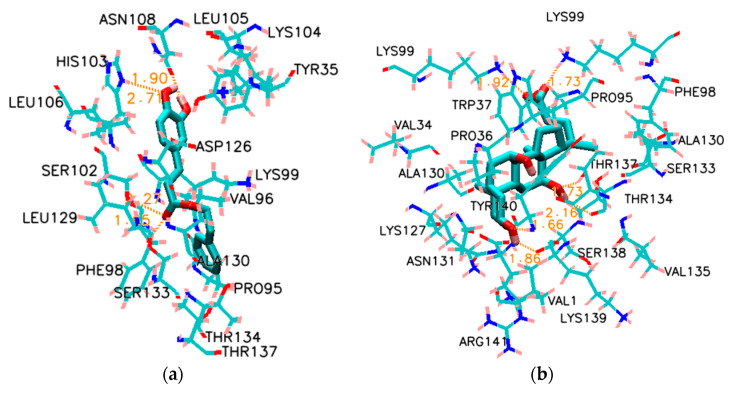
(**a**) CAPE and its surrounding amino acid residues for CAPE–Hb system; (**b**) NaC and its surrounding amino acid residues for NaC–Hb system; (**c**) NaDC and its surrounding amino acid residues for NaDC–Hb system; (**d**) The whole simulation diagram of Hb docking with CAPE, NaC and NaDC; (**e**) Partial enlargement of (**d**). The spiral is the carbonmonoxy liganded bovine Hb (PDB: 1G09 at pH 7.2), the yellow rod is CAPE, the green rod is NaC and the orange rod is NaDC.

**Table 1 molecules-28-03440-t001:** Quenching constants, binding constants and thermodynamic parameters of the interaction between CAPE and Hb in pH 7.4 phosphate buffer solution at different temperatures.

*T*	*K_sv_* (10^4^, M^−1^)	*K_q_* (10^12^, M^−1^·s^−1^)	*K_a_* (10^3^, M^−1^)	*n*	∆*G*_m_ (kJ/mol)	∆*H*_m_ (kJ/mol)	∆*S*_m_ [J/(mol·K)]
298.2	6.41 ± 0.008	6.41 ± 0.008	5.89 ± 0.013	0.78 ± 0.009	−21.5 ± 0.054	−40.8 ± 0.05	−65.1 ± 0.02
302.2	6.00 ± 0.002	6.00 ± 0.002	4.27 ± 0.009	0.76 ± 0.008	−21.0 ± 0.052		
306.2	5.81 ± 0.005	5.81 ± 0.005	3.52 ± 0.008	0.75 ± 0.014	−20.8 ± 0.057		
310.2	5.45 ± 0.003	5.45 ± 0.003	3.13 ± 0.007	0.74 ± 0.019	−20.7 ± 0.050		

**Table 2 molecules-28-03440-t002:** Fluorescence lifetimes at different molar ratios of Hb and CAPE at 298.2 K and in pH 7.4 phosphate buffer.

Molar Ratio of Hb to CAPE	$\tau_{1}$ (ns)	$\alpha_{1}$	$\tau_{2}$ (ns)	$\alpha_{2}$	τ (ns)
1:0	0.98	0.89	8.74	0.11	1.83
1:1	0.85	0.84	6.91	0.15	1.81
1:2	0.86	0.83	6.64	0.17	1.84
1:3	0.85	0.83	6.52	0.17	1.82

**Table 3 molecules-28-03440-t003:** Quenching constants and binding constants of the interaction between Hb and CAPE in the presence of NaC or NaDC in pH 7.4 buffer solution at different surfactant concentrations.

*SAA*	*C* (mM)	*K_sv_* (10^4^, M^−1^)	*K_q_* (10^12^, M^−1^·s^−1^)	*K_a_* (10^4^, M^−1^)	*n*
NaC	1.0	9.17 ± 0.014	9.17 ± 0.014	0.84 ± 0.018	0.78 ± 0.039
	3.3	7.68 ± 0.001	7.68 ± 0.001	4.47 ± 0.027	1.09 ± 0.053
	10.0	6.60 ± 0.018	6.60 ± 0.018	1.32 ± 0.015	0.84 ± 0.036
	20.0	4.51 ± 0.009	4.51 ± 0.009	0.45 ± 0.003	0.78 ± 0.036
NaDC	1.0	8.92 ± 0.016	8.92 ± 0.016	9.72 ± 0.015	0.90 ± 0.034
	2.5	7.26 ± 0.001	7.26 ± 0.001	15.84 ± 0.016	0.94 ± 0.031
	8.0	5.93 ± 0.005	5.93 ± 0.005	36.47 ± 0.065	1.19 ± 0.036
	20.0	5.39 ± 0.008	5.39 ± 0.008	10.6 ± 0.025	1.06 ± 0.039

**Table 4 molecules-28-03440-t004:** Quenching constants, binding constants and thermodynamic constants of the interaction between Hb and CAPE in the presence of NaC or NaDC in pH 7.4 buffer solution at different temperatures.

*SAA*	*T* (K)	*K_sv_* (10^4^, M^−1^)	*K_a_* (10^4^, M^−1^)	*n*	∆*G*_m_ (kJ/mol)	∆*H*_m_ (kJ/mol)	∆*S*_m_ [J/(mol·K)]
NaC	298.2	7.68 ± 0.001	4.47 ± 0.027	1.09 ± 0.053	−26.5 ± 0.008	−90.1 ± 0.499	−213 ± 0.165
	302.2	7.52 ± 0.001	3.89 ± 0.023	1.03 ± 0.044	−26.5 ± 0.008		
	306.2	7.45 ± 0.001	1.62 ± 0.021	1.08 ± 0.051	−24.7 ± 0.014		
NaDC	298.2	7.26 ± 0.001	15.84 ± 0.016	0.94 ± 0.031	−29.7 ± 0.004	−73.5 ± 0.244	−148 ± 0.809
	302.2	7.17 ± 0.002	8.51 ± 0.013	0.97 ± 0.025	−28.5 ± 0.006		
	306.2	6.59 ± 0.002	7.08 ± 0.091	0.86 ± 0.017	−28.4 ± 0.007		

**Table 5 molecules-28-03440-t005:** Binding constants of the interaction between CAPE and Hb in the presence or absence of metal ions in 298.2 K and pH 7.4 buffer solution.

Metal Ions	*K_a_* (M^−1^)	R^2^
without	5.89 × 10^3^	0.95
Ca^2+^	9.42 × 10^0^	0.99
Cu^2+^	5.88 × 10^4^	0.98
K^+^	1.46 × 10^3^	0.97
Mn^2+^	2.14 × 10^3^	0.98
Ni^2+^	7.24 × 10^3^	0.98

**Table 6 molecules-28-03440-t006:** Details of the chemicals used in the current work.

Chemical	Source	CAS Number	Storage Method	Mass Fraction Purity ^a^
CAPE	Ark Pharm, Inc., Shanghai, China	104594-70-9	Keep in dark place, Sealed in dry, Store in freezer, under 20 °C	≥0.980
Hb	Aladdin Biochemical Technology Co., Ltd., Shanghai, China	9008-02-0	Store in the freezer at 2 to 8 °C	≥0.980
NaC	InnoChem Science & Technology Co., Ltd., Beijing, China	361-09-1	Stored in desiccator	≥0.990
NaDC		302-95-4		≥0.990
Na_2_HPO_4_	Damao Chemical Reagent Co., Ltd., Tianjin, China	7558-79-4		≥0.990
NaH_2_PO_4_		7558-80-7		≥0.990
CaCl_2_		10043-52-4		≥0.996
KCl		7447-40-7		≥0.990
CuCl_2_·2H_2_O		1344-67-8		≥0.990
MnCl_2_·4H_2_O		13446-34-9		≥0.990
NiCl_2_·6H_2_O		7791-20-0		≥0.980
C_2_H_5_OH		64-17-5	Stored in dark place, sealed in dry	≥0.997

^a^: As stated by the supplier.

## Data Availability

The data presented in this study are available upon request from the corresponding author.

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
