# Peer review of "Effects of Temperature, Metal Ions and Biosurfactants on Interaction Mechanism between Caffeic Acid Phenethyl Ester and Hemoglobin"

_molecules, 2023, doi:10.3390/molecules28083440_

Round 1

Reviewer 1 Report (Previous Reviewer 4)

The manuscript could be accepted for publication after these improvements.

Author Response

Reply: We feel very grateful for your evaluating our research and we are really grateful for the reviewer’s affirmation of our manuscript. Thanks you very much.

Reviewer 2 Report (New Reviewer)

Comments on molecules-2266243

In this study, the author has studied “Effects of Temperature, Metal Ions and Biosurfactants on Interaction Mechanism between Caffeic Acid Phenethyl Ester and Hemoglobin.This is an engaging article with a robust approach that purposefully questions our knowledge of the subject. However, the presentation of the results is somewhat confusing, and the readability of the discussion could be improved. The discussion section needs to be critically revised. Some figures need to be labeled. The English language used in the manuscript needs some improvements as some punctuation, and grammatical mistakes are present. Experimental designs required more clarity. Moreover, research models are not discussed in an understandable manner, reflecting that the author needs a more comprehensive way of thinking.

Specific comments:

1.      The abstract needs to be critically revised. Please add clear results and the conclusion of the study. The objective of the study is also unclear.

2.      Please add more strong keywords and avoid the words used in the title.

3.      There is no need for a figure in the introduction section. Please remove it.

4.      The research gap is unclear in the present study. Please add some data in the introduction about factors affecting the interaction mechanism between CAPE and HB. It is also suggested to add some literature on the mechanism of CAPE.

5.      What is the research gap and novelty of the present study?

6.      Please label Figure 2a. What is the representation of different colors?

7.      Page 2, line 82: Please add the sample size and also draw a flow chart including patients with inclusion and exclusion criteria.

8.      What was the optimum temperature on the binding interaction of CAPE-Hb?

9.      Please add discussion in “UV-vis spectrum.”

10.  Please label Figure 3a. What is the representation of different colors?

11.  Please add discussion in “Effect of biosurfactants (NaC, NaDC) on the binding interaction of CAPE-Hb.”

12.  The discussion needs to be revised. The discussion needs professional English editing, and please revise them carefully to make it standard. Please focus on the main topic during the discussion. An excellent discussion contains an accurate statement of the results, the relevance, and importance of the results, suitable comparisons to similar studies, alternative explanations of the findings, known limitations, and suggestions for future research.

13.  The authors could add a heading of “Statistical analysis” at the end of the methodology.

14.  Authors are advised to proofread the manuscript to overcome grammatical mistakes.

15.  Authors are advised to revise headings and subheadings.

Author Response

Reply: We thank the reviewer for evaluating our research and we are really grateful for the reviewer’s affirmation of our manuscript. You advice are very helpful for us. We have polished the language. We have revised the manuscript in accordance with the suggestion of the Reviewer. The changed places are marked with red and blue colors. The detailed changes are listed below.

Reviewer 3 Report (New Reviewer)

Overall, the manuscript is scientifically incomplete and/or lacks a significant, novel contribution to the field.

The authors have recently reported on the interactions of CAPE with trypsin, pepsin and α-chymotrypsin (see Ref. 8) and with bovine serum albumin (see Ref. 7).

Overall, the experimental approach does not adequately or convincingly answer the posed questions.

Therefore, the paper does not deserves to be published in Molecules.

Mayor issues:

a) The authors seems to be not aware that the concentrations being used in the reported spectroscopic characterizations are above the Critical Micelle Concentrations of NaC and NaDC, respectively.

b) The authors did not provide computational details nor results  of their molecular dynamics simulations. Instead, they only report on the molecular docking of Hemoglobin with CAPE, NaC and NaDC, respectively.  The authors have used as a template for the molecular docking the crystal structure of deoxy human hemoglobin (R-state)  bound to the strong effector L35 (PDB 2D60).

The authors should have performed a comparative molecular docking using as templates also the available crystals structures of human oxy-, carboxy-, and carbonmonoxy- human hemoglobins (R- and T-states).

The authors should have thoroughly investigated the effects of CAPE and/or NaC and NaDC on the human deoxy- oxy-, carboxy-, and carbonmonoxy- human hemoglobins.

Author Response

Reply: We thank the reviewer for willing to spend time to evaluate our research, although you give the negative comments. We think that it has some meaning about the interaction between CAPE and Hb, especially the results from the effect of metal ions and biosurfactants on the interaction. We still revised the manuscript as you suggested and the other reviewers’ comments. Because the other reviewers gave the positive comments to the manuscript, so we have the chance to revise and polish it. Would you like to reevaluate the manuscript and give us the positive comment and the encouragement? Whatever the result is, we must make great efforts anyway.

Round 2

Reviewer 2 Report (New Reviewer)

The authors have carefully addressed all the comments. So, the manuscript should be accepted in its present form.

Author Response

Reply: We feel very grateful for your evaluating our research and we are really grateful for the reviewer’s affirmation of our manuscript. Thanks you very much.

Reviewer 3 Report (New Reviewer)

The authors have satisfactorily addressed only in part my concerns, i.e. the effects of NaC and NaDC concentrations respectively.

In the Introduction (lines 87-88) the authors state "In addition, molecular dynamic simulation can offer theoretical evidence for the type of force between Hb and CAPE."

Unfortunately, the authors do not report on Molecular Dynamics simulations but only on Molecular Docking. 

It is not clear why the authors have choosen as template the carbonmonoxy liganded bovine hemoglobin (PDB 1G09) for their the Molecular docking. In the original manuscript, the authors have used as a template for the molecular docking the crystal structure of deoxy human hemoglobin (R-state) bound to the strong effector L35 (PDB 2D60).

As previously pointed out, the authors should have performed a comparative molecular docking using as templates also the available crystals structures of human oxy-, carboxy-, and carbonmonoxy- human hemoglobins (R- and T-states). The authors should have thoroughly investigated the effects of CAPE and/or NaC and NaDC on the human deoxy- oxy-, carboxy-, and carbonmonoxy- human hemoglobins.

Author Response

Reply: We thank the reviewer for willing to spend time to evaluate our research.

Sorry, we confuse the concept of molecular docking and molecular dynamics simulation, and we have changed “molecular dynamics simulation” to “molecular docking” in the lines 87-88.

Because we used the bovine hemoglobin in the experiments, we chose the bovine hemoglobin (PDB 1G09) as template for the molecular docking. In the last revision, we chose the human hemoglobin as template for the molecular docking. We think that the bovine hemoglobin is more suitable than the human hemoglobin due to the connection between experimental and theoretical results.  The part of the human hemoglobin can be found in the pdf file.

Round 3

Reviewer 3 Report (New Reviewer)

The authors are kindly requested to perform a comparative molecular docking analysis by using as templates besides the carbonmonoxy liganded bovine Hb (PDB 1G09) also the available crystals structures of  bovine deoxy Hb (PDB 1HDA) and the CO form of bovine Hb (PDB ID 6ll1) .

It is not clear which form (and state) of bovine Hb has been used by the authors in their experiments.

The authors should have thoroughly investigated the effects of CAPE and/or NaC and NaDC on deoxy-, oxy-, carboxy-, and carbonmonoxy- bovine Hb forms ((R- and T-states).

Furthermore, the overall effect of the pH should have been taken into account, as well.

Author Response

We thank the reviewer for willing to spend time to evaluate our research. We have added the molecular docking part as suggested.

This manuscript is a resubmission of an earlier submission. The following is a list of the peer review reports and author responses from that submission.

Round 1

Reviewer 1 Report

In this submitted paper, the authors tried to investigate the effects of temperature, metal ions and biosurfactants on the interaction between caffeic acid phenylethyl ester (CAPE) and hemoglobin (Hb) by some spectroscopic methods and molecular docking calculations. The manuscript can be accepted for publication in the Molecules. However, some major issues underlined in the following comments need to be resolved before further consideration.

Major issues:

1.      P.1, L.26, it would be a good idea to show the chemical structure of CAPE in an illustration.

2.      Did the authors take into account the inner filter effect (IFE) during performing fluorescence experiments?

3.      For a better understanding of the contribution of Trp or Tyr residues in the fluorescence quenching process, the authors need to conduct another fluorescent-based method, synchronous fluorescence, in their experiments.

4.      For determining the effects of the ligand studied here, the expansion of the UV scanning wavelength below 250 nm seems to be necessary.

5. Figure 5, the position of the ligand docked on the whole protein Hb has to be shown in an extra image.

Author Response

Replies to the comments of the Reviewer

Reviewer #1

In this submitted paper, the authors tried to investigate the effects of temperature, metal ions and biosurfactants on the interaction between caffeic acid phenylethyl ester (CAPE) and hemoglobin (Hb) by some spectroscopic methods and molecular docking calculations. The manuscript can be accepted for publication in the Molecules. However, some major issues underlined in the following comments need to be resolved before further consideration.

Major issues:

Reply: We thank the reviewer for evaluating our research and we are really grateful for the reviewer’s affirmation of our manuscript. We have revised the manuscript in accordance with the suggestion of the Reviewer. The detailed changes are listed below.

Comments:

  • 1, L.26, it would be a good idea to show the chemical structure of CAPE in an illustration.
  • Reply: We are very grateful for this suggestion. We have added the chemical structure of CAPE in P.2.
  • Did the authors take into account the inner filter effect(IFE) during performing fluorescence experiments?
  • Reply: Thanks for your suggestion. As you mentioned, the inner-filter effect is not negligible.We are very sorry for the lack of explanation for our treatment of inner-filter effect in the article. It has now been included in the text as follows, “Considering the inner filter effects of CAPE on the fluorescence spectra of Hb, the fluorescence intensities in this experiment were corrected by the following equation

                 (2)

where Fcor, Fobs, Aex, and Aem are the corrected fluorescence intensity, the observed fluorescence intensity, and the absorbance values of the sample at excitation and emission wavelengths, respectively”.

  • For a better understanding of the contribution of Trp or Tyr residues in the fluorescence quenching process, the authors need to conduct another fluorescent-based method, synchronous fluorescence, in their experiments.
  • Reply:We are very grateful for this suggestion. The synchronous fluorescence spectra of âˆ†λ= 15 nm and ∆λ= 60 nm can often used to show the microenvironment changes around Tyr and Trp residue, and UV-vis spectroscopy can also be used to detect the microenvironment changes of them. We had used UV-vis spectroscopy to confirm the microenvironment changes, so we had not conducted synchronous fluorescence method. If needed, please tell us and we will supply the synchronous fluorescence experiments in the next revised process.
  • For determining the effects of the ligand studied here, the expansion of the UV scanning wavelength below 250 nm seems to be necessary.
  • Reply: Thank you for your suggestion. We have expanded the X-axis from 200 nm to 450 nm. The left is the revised figure 3a and the right is the raw figure 3a.
  • Figure 5, the position of the ligand docked on the whole protein Hb has to be shown in an extra image.
  • Reply:Thank you for your suggestion. We have placed the whole image in the supporting material. Figure S1 Simulation diagrams of molecular docking between Hb and CAPE/NaC/NaDC; Figure S2 Simulation diagram of Hb docking with CAPE, NaC and NaDC, in one diagram.

Reviewer 2 Report

The manuscript "Effects of Temperature, Metal Ions and Biosurfactants on Interaction Mechanism between Caffeic Acid Phenethyl Ester and Hemoglobin" submitted by Li et. al. describes the of study  of interaction of Caffeic acid phenylethyl ester (CAPE) with Hemoglobin using various biochemical methods like UV-Vis spectroscopy, Fluorescence spectroscopy, CD, DLS and Molecular docking. This study lacks originality. Moreover, the significance of the study and its application are not properly mentioned. This study is nothing but some routine works. Similar interaction studies of  CAPE and BSA has been published. 

To me, this manuscript does not qualify to be recommended for publication in the reputed journal like Molecules.

Author Response

Replies to the comments of the Reviewer

Reviewer #2: 

Reply: We thank the reviewer for willing to spend time to evaluate our research. We are very sorry about lacking originality. Propolis is very popular as a health food, and as the biologically active constituent of it, CAPE is of great concern due to its efficacy in human health and diseases. So the mechanism of the constituent of propolis and proteins is very meaningful to promote the natural food. Indeed, the interaction studies of CAPE and BSA had been published, but we still think that it has some meaning about the interaction between CAPE and Hb, especially the results from the effect of metal ions on the interaction. About one month ago, I had been told that the journal of “Molecules” will publish a special issue about “Recent Advances in Polyphenol Compounds”. We are very glad that our work is very suit for this issue. We have revised the manuscript point by point based on the reviewers’ comments. Would you like to reevaluate the manuscript and give some advice to us to improve the manuscript? Thank you for evaluating our research again.

Reviewer 3 Report

Dear authors,

I reviewed the manuscript and it seems very interesting to decipher how hemoglobin interacts with other molecules, specifically with drugs of natural origin, which are still in the experimental stage. I believe that the work could be published after making some minor corrections, for which I have the following comments.

L41. Write kDa instead of kD

L 49-52. For the benefit of non-specialist readers, I suggest rewriting this sentence: "First, bile salts (BSs) are naturally occurring amphiphilic substances that are synthesized and released by the liver and stored in the gallbladder, and sodium deoxycholate (NaDC) and sodium cholate (NaC) are primarily considered in the study of the interaction between Hb and ligand". In its current form it is not very clear why NaDC and NaC are the preferred compounds to study the Hb-Ligand interaction. In fact, in section 3.2 (L 270) they are defined as biosurfactants, but I think it should be made explicit from the start.

L 86-88. The phrase "Take 200 µL of 100 µM Hb stock solution in a test tube, add a given volume of 400 µM CAPE stock solution, then dilute to 4 mL with 10 mM pH 7.4 phosphate buffer and obtain the different CAPE concentrations desired", sounds like as if reading the protocol from a handbook. Re-write according to the style of a scientific report.

L 89. Write min instead of minutes

I recommend avoiding that data presented in tables is also presented in the text, for example as was done in L 215-217.

L372-374. What does it imply, within the context of the work, that these binding energy values have been obtained? It seems as if the discussion is left unfinished.

In materials and methods nothing is mentioned about the experimental design or statistical analysis. How many times was the experiment repeated?

In general the English is well written and understandable, but I think it could be polished.

Author Response

Replies to the Reviewer

Reviewer #3

Dear authors,

I reviewed the manuscript and it seems very interesting to decipher how hemoglobin interacts with other molecules, specifically with drugs of natural origin, which are still in the experimental stage. I believe that the work could be published after making some minor corrections, for which I have the following comments.

Reply: We thank the reviewer for evaluating our research and we are really grateful for the reviewer’s affirmation of our manuscript. We have revised the manuscript in accordance with the suggestion of the Reviewer. The detailed changes are listed below.

L41. Write kDa instead of kD

  • Reply:Now in 2, L.54, "kD" changed to "kDa"

L 49-52. For the benefit of non-specialist readers, I suggest rewriting this sentence: "First, bile salts (BSs) are naturally occurring amphiphilic substances that are synthesized and released by the liver and stored in the gallbladder, and sodium deoxycholate (NaDC) and sodium cholate (NaC) are primarily considered in the study of the interaction between Hb and ligand". In its current form it is not very clear why NaDC and NaC are the preferred compounds to study the Hb-Ligand interaction. In fact, in section 3.2 (L 270) they are defined as biosurfactants, but I think it should be made explicit from the start.

  • Reply:We are very grateful for this suggestion. We changed this part as "First, biosurfactant may alter the binding behavior of drug on protein to regulate the drug release, thus the biosurfactants of bile salt (BSs), including sodium deoxycholate (NaDC) and sodium cholate (NaC), were chosen as the influencing factor in this study. Bile salts (BSs) are naturally occurring amphiphilic substances that are synthesized and released by the liver and stored in the gallbladder, which are often used to study their effect on the interaction between protein and ligand."

L 86-88. The phrase "Take 200 µL of 100 µM Hb stock solution in a test tube, add a given volume of 400 µM CAPE stock solution, then dilute to 4 mL with 10 mM pH 7.4 phosphate buffer and obtain the different CAPE concentrations desired", sounds like as if reading the protocol from a handbook. Re-write according to the style of a scientific report.

  • Reply:Thanks for your suggestion. We changed this part as "The proper amount of the stock solutions, Hb (100 µM) and CAPE (400 µM), were taken and were diluted to the desired concentrations with phosphate buffer."

L 89. Write min instead of minutes

  • Reply:Now in 11, L.364, "minutes" changed to "min".

I recommend avoiding that data presented in tables is also presented in the text, for example as was done in L 215-217.

  • Reply:We delete the sentences in L. 215-217, and changed it as "The fluorescence lifetime τ of Hb did not change significantly with CAPE addition or its concentration."

L372-374. What does it imply, within the context of the work, that these binding energy values have been obtained? It seems as if the discussion is left unfinished.

  • Reply:This suggestion is very valuable for verifying the fluorescence spectroscopy with molecular docking. We added the sentences as "This showed that the combinations were spontaneous and the binding tendency of biosurfactant on Hb was stronger than CAPE. The different binding position of biosurfacant and CAPE was help to verify the result of fluorescence spectroscopy. The strongly combination of surfactant (NaC, NaDC) and Hb induced the changed tertiary structure of the protein. But the existence of surfactant did not occupy the position of CAPE, on the contrary, the binding interaction between CAPE and Hb was strengthened."

In materials and methods nothing is mentioned about the experimental design or statistical analysis. How many times was the experiment repeated?

  • Reply:Now in 11, L.375, we added "The results of three scans were averaged and background fluorescence was corrected."

In general the English is well written and understandable, but I think it could be polished.

  • Reply:Sorry, this is our shortcomings. We revised the English again with the help of one reviewer.

Reviewer 4 Report

Li et al. studied about theEffects of Temperature, Metal Ions and Biosurfactants on Interaction Mechanism between Caffeic Acid Phenethyl Ester and  HemoglobinIt is a good approach but the paper can be accepted for publication only after minor revision. It is recommended that the following aspects be addressed by the authors.

1/Introduction:  authors reported the multiple biological activities of CAPE (antibacterial, antiviral, anti-inflammatory, antioxidative, antiplatelet, and antitumor) but they did not explain the mechanisms that support or explain these activities (specific, nonspecific, targets, etc.). I suggest adding a paragraph to briefly describe these mechanisms.

2/MM:  the choice of the metals should be argued along with their chosen concentrations.

*To study the interactions between CAPE and HB, authors used several techniques such as ultraviolet-visible spectroscopy (UV-vis), fluorescence spectroscopy, circular dichroism (CD), dynamic light scattering (DLS), and molecular docking (in silico approach) using an “artificial environment” mimicking the real conditions. I am not sure that these conditions could really be similar to real ones to be able to take a conclusion regarding the interaction between Hb and CAPE. In other words, why authors could not use blood samples to conduct the study?

3/Resuts and discussion: This section is well written and presented, I have a remark regarding the docking analysis. Why it was not completed with dynamic simulation or other in silico studies?

4/Conclusion: I want to see the conclusion regarding the impact of this interaction on the possible use of CAPE as a drug after conducting this study. 

Author Response

Replies to the comments of the Reviewer

Reviewer #4: 

Li et al. studied about the “Effects of Temperature, Metal Ions and Biosurfactants on Interaction Mechanism between Caffeic Acid Phenethyl Ester and  Hemoglobin” It is a good approach but the paper can be accepted for publication only after minor revision. It is recommended that the following aspects be addressed by the authors.

Reply: We thank the reviewer for providing constructive feedback and we are grateful for the reviewer’s affirmation of our manuscript. We have fully revised the manuscript based on the reviewer’s comments and replied to each item. The detailed changes are listed as below.

1/Introduction: authors reported the multiple biological activities of CAPE (antibacterial, antiviral, anti-inflammatory, antioxidative, antiplatelet, and antitumor) but they did not explain the mechanisms that support or explain these activities (specific, nonspecific, targets, etc.). I suggest adding a paragraph to briefly describe these mechanisms.

  • Reply: Thank you for your suggestions. We added several sentences about the acting mechanism of CAPE as follows “It was reported that CAPE maydirectly target the signaling pathways related to oxidation, inflammation and cancer development, such as NF-κB (nuclear factor-κB), MAPK (mitogen-activated protein kinase), PI3K/PKB (phosphoinositide 3-kinase/protein kinase B) and NLPR3 (NOD-like receptor thermal protein domain associated protein 3), then combat the oxidative stress, inflammation, proliferation, migration and invasion of cells [3,4,6]. Thus CAPE becomes an important target drug for the prevention and treatment of cardiovascular, neurological and cancer-related diseases. However, the current research results of CAPE are mainly from cellular and animal model experiments. If the researches are from the perspective of pharmacodynamics and pharmacokinetic aspects, it can promote the further development and clinical transformation of CAPE”.

2/MM: the choice of the metals should be argued along with their chosen concentrations.

  • Reply: Thank you for your suggestion, it is very worth thinking about. The focus of this part is the effect of different metal ions on the interaction, so we fixed the concentration of them. Through literature reading, we found that the metal ion concentration was fixed at around 100 μM. Considering that several chosen metal ions are the trace elements, we had chosen the concentration of the ions at 60 μ Through this comment, we find that it is important to discuss the concentration of metal ions influencing the interaction between CAPE and Hb. Tremendous amount of work involved in discussing this problem is necessary, and we will consider it in the next step. Cu2+, Ca2+, K+, Mn2+or Ni2+ are the essential life elements in the human body and plays important structural and functional roles in many proteins, especially the ions of Cu2+ and Ca2+. And we discussed them related to the binding constant. Now for this problem, we added the sentences as follows “These metal ions are the essential life elements involved in various human metabolism processes, whose concentrations were fixed at 60 μM considering their trace existence in the human body”.

*To study the interactions between CAPE and HB, authors used several techniques such as ultraviolet-visible spectroscopy (UV-vis), fluorescence spectroscopy, circular dichroism (CD), dynamic light scattering (DLS), and molecular docking (in silico approach) using an “artificial environment” mimicking the real conditions. I am not sure that these conditions could really be similar to real ones to be able to take a conclusion regarding the interaction between Hb and CAPE. In other words, why authors could not use blood samples to conduct the study?

  • Reply: The current research of us is limited in the in-vitroresearch, and phosphate buffer solution was often used here. There has strict regulatory control for blood samples, so the selective conditions are expected to the further study using blood samples.

3/Resuts and discussion: This section is well written and presented, I have a remark regarding the docking analysis. Why it was not completed with dynamic simulation or other in silico studies?

  • Reply: Because we thought that it was enough to explain the problem using the molecular docking method, so we didn’t use dynamic simulation. Usually, ifthe docking results are not satisfactory, we will consider the dynamic simulation method.

4/Conclusion: I want to see the conclusion regarding the impact of this interaction on the possible use of CAPE as a drug after conducting this study.

  • Reply: Your suggestionsis very good, and it is very helpful to improve and enrich the conclusion part. We have added the sentences as follows “These studies are significant as the structure of Hb can be effectively adjusted by the presence of CAPE under different environment. This adjustment might be relevant to targeted drug delivery and release for the injection and oral medication, when CAPE is used for treatment of cardiovascular, neurological and cancer-related diseases. Therefore, the binding study of drugs to Hb is greatly important in pharmacology, pharmacokinetics and so on. Besides, these results provide a theoretical basis and research basis for the studies of the properties of CAPE, sustained release in vivo and anti-tumor mechanism, etc.”

Reviewer 5 Report

Dear Authors,

Please find attached my comments and suggestion.

I recommend using antiplagiarism software to check text originality.

However, after Plagiarism checking (Turnitin), I recommend suggesting to the authors a careful improvement of Molecular docking (2.8) and  Fluorescence quenching spectrum (3.1.1.). After scanning I got 40% with and 31% without considering references.

Kind regards,

Reviewer 1

Author Response

Replies to the Reviewer

Reviewer #5

Please find attached my comments and suggestion.

I recommend using antiplagiarism software to check text originality.

However, after Plagiarism checking (Turnitin), I recommend suggesting to the authors a careful improvement of Molecular docking (2.8) and Fluorescence quenching spectrum (3.1.1.). After scanning I got 40% with and 31% without considering references.

Reply: We thank the reviewer for evaluating our research and we are really grateful for the reviewer’s affirmation of our manuscript. The reviewer’s work is very meticulous, and we have revised the manuscript in accordance with the suggestion of the Reviewer. The detailed changes are listed below.

For the fluorescent part, it is may be that the parameters of the formulas needed to be defined and explained, and induced the high repetition rate. It it is not allowed, we will be glad to put the relevant formulas into the supporting material.

For the part of Molecular docking (3.8 now), we have deleted some sentences as “The water molecules of Hb were removed during docking, and polar hydrogen atoms were added. During the molecular docking process, the grid size of drug and protein was 126 Å×126 Å×126 Å, and the grid spacing was 1 Å. During docking, drug and protein are randomly docked 100 times and calculated with AutoGrid 4.0 software”, added the sentence as “The relative information about the molecular docking process, please see the ref.”

The other changes are listed below:

P.1, L.19., "force" changed to "the type of the main interaction force".

P.1, L.21, "bond" changed to "bonds", "force" changed to "forces".

P.2, L.54, "kD" changed to "kDa".

P.2, L.78, "Fluorescence quenching spectrum" changed to "Fluorescence quenching".

P.3, L.94, "presence or absence" changed to "absence or presence".

P.3, L.133, "formula" changed to "the equation".

P.3, L.134, "is" changed to "was".

P.4, L.141, the ordinate of Figure 2(a)"Fluorescence intensity" changed to " Fluorescence intensity(a.u.)".

P.4, L.145, modified the format of "µM".

P.4, L.147, "and" changed to "versus".

P.4, Table 2: "-21.51" changed to "-21.5", "-20.99" changed to "-21.0", "-20.78" changed to "-20.8", "-20.74" changed to "-20.7", "-40.76" changed to "-40.8", "-65.07" changed to "-65.1". 

P.5, L.158, added definitions of "τ" and "α". Where τ is the average fluorescence lifetime, α1 and α2 are the pre-exponential factors for the first and second decay times τ1 and τ2.

P.5, L.165, "is" changed to "are".

P.5, L.175, "Hb" changed to "heme moiety".

P.5, L.178, "structure" changed to "tertiary structure".

P.5, L.180, "This indicates that the interaction between CAPE and Hb does not destroy the stability of heme in Hb" changed to "It suggests that the stability of heme is not significantly affected, although the interaction between CAPE and Hb induces the pocket conformation change of the protein confirmed by the fluorescence method and the subsequent molecular docking".

P.6, L.197, "This is" changed to "These data are".

P.7, L.217, the ordinate of Figure 3(c)"Intensity" changed to "Intensity (%)", the abscissa of Figure 3(c) "Size(d.nm)" changed to "Size(nm)"

P.7, L.230, "enhanced" changed to "decreased".

P.8, L.257, the ordinate of Figure 4(b)"Fluorescence intensity" changed to "Fluorescence intensity(a.u.)".

P.9, L.259, the ordinate of Figure 4(c)"Fluorescence intensity" changed to "Fluorescence intensity(a.u.)".

P.9, L.263, modified the format of "µM".

P.9, Table 3: "-26.53" changed to "-26.5", "-26.54" changed to "-26.5", "-24.66" changed to "-24.7", "-29.66" changed to "-29.7", "-28.50" changed to "-28.5", "-28.41" changed to "-28.4", "-90.11" changed to "-90.1", "-73.49" changed to "-73.5", "-212.71" changed to "-213", "-147.91" changed to "-148",

P.10, L.297, "metal ions" changed to "five different metal ions".

P.12, L.331, The left is the revised figure 6 and the right is the raw figure 6.

P.13, L.354, "quality" changed to "quantity".

P.13, L.359, "The concentration" changed to "The final concentration".

P.13, L.365, "200nm" changed to "290nm". The scan starts at 200nm, but the data before 290nm is useless, so we change it to start at 290nm.

P.14, L.379, "the special software" changed to "the Fluoracle software".

P.14, L.399, "ZETASIZERN NANO software" changed to "ZETASIZER NANO software".

P.15, L.413, "drugs" changed to "the transport of drugs".

P.15, L.431, "mechanism, etc." changed to "mechanism etc.".

P.16, L.493, "563" changed to "563-570".

P.16, L.518, "adv" changed to "Adv".

Round 2

Reviewer 1 Report

The authors have performed all the comments recommended by the reviewer. Thus, the revised version of the manuscript could be accepted for publication in Molecules.

Reviewer 2 Report

The revised manuscript does show any improvement as it lacks the originality. To me it is a routine work with no novelty.  I am sorry, I decide to stick  with my original decision to reject the manuscript.